# Efficient, Formal, Material, and Final Causes in Biology and Technology

**DOI:** 10.3390/e25091301

**Published:** 2023-09-05

**Authors:** George F. R. Ellis

**Affiliations:** Mathematics Department, The New Institute, University of Cape Town, 20354 Hamburg, Germany; george.ellis@uct.ac.za

**Keywords:** Aristotle, causation, emergence, causal closure, downwards causation

## Abstract

This paper considers how a classification of causal effects as comprising efficient, formal, material, and final causation can provide a useful understanding of how emergence takes place in biology and technology, with formal, material, and final causation all including cases of downward causation; they each occur in both synchronic and diachronic forms. Taken together, they underlie why all emergent levels in the hierarchy of emergence have causal powers (which is Noble’s principle of biological relativity) and so why causal closure only occurs when the upwards and downwards interactions between all emergent levels are taken into account, contra to claims that some underlying physics level is by itself causality complete. A key feature is that stochasticity at the molecular level plays an important role in enabling agency to emerge, underlying the possibility of final causation occurring in these contexts.

## 1. Introduction

Fully understanding causation and fully explaining why complex systems are the way they are and behave the way they do requires holistic, historical, contextual, and extended views of causation across levels. As Aristotle was the first person we know of to articulate this holistic view, it is worth exploring whether, taking into account scientific discoveries made since his time, revisiting his understandings might be useful to us today. I suggest here that useful light is thrown on the way that life emerges from the underlying physics and molecular biology by considering updated versions of his four types of causation—efficient, formal, material, and final (Shields 2022 [1])—and that the same is true for understanding the emergence of technology (Arthur 2009 [2]), specifically digital computers. This argument gives strong support to an emergentist position in biology and technology.

This paper is not a disquisition on Aristotle’s views as such. Rather, it uses the basic notions of his four kinds of causation as a foundation for providing a useful, fine-grained view of the nature of causation in these contexts. The paper is also not a discussion of the contentious issue of whether any variety of strong or weak emergence occurs. The concern is how emergence that occurs in these contexts is characterized by genuinely effective laws relating emergent concepts and variables without considering the logically separate issue of whether this emergent higher-level behavior can be uniquely predicted from lower-level elements and laws alone. 

The paper also does not include a discussion of causation in the cosmological context, where the nature of final causation becomes a controversial issue.

What this paper does do, is firstly, claim that in the specific cases of concern, there are verifiable same-level laws of behavior characterizing causation at every emergent level, specified in terms of variables applicable at that level. This is shown to be true by providing specific examples from various contexts where it indeed occurs, as can be demonstrated by suitable experiments. Existence of these effective laws is how the different emergent levels (and associated academic disciplines) are identified. Why are they considered effective laws? Because whatever their origin, they reliably determine outcomes given suitable initial data and boundary conditions. This may be claimed to be a useful updated concept of efficient causation. How does efficient causation arise? By coarse-graining lower-level variables and dynamics (i.e., viewing them at a larger scale) or black-boxing lower-level logical relations. This results in both formal and material causation occurring at the relevant emergent level, and possibly final causation as well; thus, these are aspects enabling efficient causation, occurring concurrently. Formal and material causation are associated with symmetry breaking. Specific outcomes are determined by setting boundary and initial conditions, including those provided by higher levels (Noble 2012 [3]). Secondly, a key point is that any of these forms of causation can take place in either synchronic or diachronic forms: they can be time independent or time dependent. Thirdly, downwards causation can take place via any of suitably generalized formal, material, and final causation, contrary to claims that downward causation cannot occur. Thus, causal closure can only occur when all levels related by upwards emergence and downwards causation are included in the set considered. No physics level (of which there are a number) is causally complete by itself.

Why is this worth doing? Not only because Aristotle gave the germ of the idea in each case, which is indeed true, and so provides a link back to long before Galileo Galilei, Isaac Newton, and other such pioneers, but because it gives a viewpoint on emergence that is more comprehensive in its ambit than most other approaches. 

This section looks at Aristotle’s four causes (Section 1.1), scientific discoveries since then (Section 1.2), the theme of updated versions of Aristotle’s causes (Section 1.3), and then gives a brief outline of this paper (Section 1.4).

### 1.1. Aristotle’s Four Causes (This account is due to Christoph Horn (Bonn). I thank him for it)

Aristotle’s account of the four causes is given in *Physics* II 3 and in *Metaphysics* V 2. Whereas our modern concept of a cause refers to an event under the influence of which a certain state of the world is brought about, Aristotle presents his general idea of causes (*aitiai*) as the answer to the why-question (*to dia ti*) in general. Aristotle’s account of causality comprises four ways to explain a given state of the world. He identifies four kinds of entities which can be used in an explanation:**The material cause**: “that out of which”, e.g., the bronze of a statue or the silver that makes a bowl.**The formal cause**: “the form”, “the account of what-it-is-to-be”, e.g., the shape of a statue.**The efficient cause**: “the primary source of the change or rest”, e.g., the artisan, the art of bronze-casting the statue, the man who gives advice, the father of the child.**The final cause**: “the end or the good, that for the sake of which a thing is done”, e.g., health is the end of walking, losing weight, purging, drugs, and surgical tools.

Causation appears in different meanings, as Aristotle claims, and it may be that all four causes appear in the explanation of an entity or a fact (*Phys*. 195 a 3–5). In the case of an artifact such as a bronze statue or a silver bowl, the bronze and the silver explain the things as their *material cause*. The material cause is directly interrelated with the *formal cause*, i.e., the shape of the statue or the bowl. Additionally, Aristotle introduces *the efficient cause* as that which is what produces the statue. He thinks that this is both the artisan and the technique of processing the material. The *final cause* is the end or the good for which the craftsman produced the artifact. Aristotle explicitly says that this “good” need not be a real good; it suffices that the agent considers something as a good. 

### 1.2. Scientific Discoveries since Then

Our understanding of the functioning of the natural and biological world is now based on two ideas: firstly, that there are some fundamental immutable laws of nature that underlie how all matter behaves, and secondly, that complex systems such as plants and animals and ecosystems and digital computers arise through very complex combinations of various kinds of particles or fields subject to those basic physical laws. 

**Basic Constituents and Laws:** Everything we see around us is composed of particles that are subject to forces that determine how they move. This is explained clearly in the famous *Feynman Lectures in Physics*, Volumes I and II (Feynman 2013 [4]). 

**Constitution of matter:** One of the most fundamental discoveries ever made is that everything is made of extremely small atoms—a fact that is far from obvious. They, in turn, are made of smaller particles (protons, neutrons, and electrons, characterized by their mass **m** and electric charge **e**), with the first two made up of even smaller particles called quarks. Atoms consist of a small nucleus comprising a number **N** of positively charged protons and an approximately equal number of uncharged neutrons, surrounded (in the neutral state) by a cloud of **N** negatively charged electrons that occur in shells at increasing distance from the nucleus (**N** is the atomic number). Combinations of atoms can form gases, liquids, glasses, crystals, or molecules.

Quantum theory (Feynman 2013 [4]: Volume III) shows that all matter has both a particulate and wave like nature; even light is made of particles (“photons”). Special relativity shows that mass **m** and energy **E** can be transformed into each other according to the famous Einstein formula **E = m c^2^**, where **c** is the speed of light.

The four forces acting on particles are gravity, electromagnetism, the weak force, and the strong force, which can be represented as fields filling space (“the gravitational field”, “the electromagnetic field”), thereby solving the puzzle as to how they can act at a distance. Electric and magnetic fields interact with each other in such a way that they can create electromagnetic waves that travel at the speed of light **c**, with a wavelength **λ** that can range from the very long (radio waves) to the very short (X-rays and γ-rays), including light itself: the part of the electromagnetic spectrum that is visible to us, with different colors corresponding to different wavelengths.

**Chemical elements:** Combining these basic components gives rise to substances of various kinds that are the link to higher-order complexity. In Aristotle’s case, these were earth, air, fire, and water, not considered as elementary in their own right but as a heap or foundation for complex possibilities (Sokolowski 1970 [5]). 

Nowadays, elementary particles are understood to give rise to atoms of various kinds that are characterized as chemical elements such as hydrogen, carbon, nitrogen, oxygen, iron, and so on. They are nature’s building blocks (Emsley 2011 [6]), with chemical properties that depend on their atomic number **N** via the column they belong to in the ***Periodic Table of the Elements*** (Scerri 2019 [7]).

**Biomolecules:** Atoms can be combined into the carbon-based organic molecules that are the foundation of life. Particularly important are macromolecules (Lehn 2004 [8]): on the one hand, nucleic acids DNA and RNA are the basis of storing and reading genetic information (Watson et al., 2014 [9]), and on the other hand, the proteins are the workhorses of molecular biology (Petsko and Ringe 2004 [10], Wagner 2014 [11]).

**Complex systems: Basic principles:** Modular hierarchical structures underlie the emergence of complexity in general, and of life in particular, for very good reasons (Booch et al., 2007 [12]), as discussed in Section 2. Through downward causation (Pezzulo and Levin 2016 [13], Vooshholz and Gabriel 2021 [14]), all emergent levels have causal powers (Noble 2012 [3]). They are adapted to their environments to greater or lesser degree by evolutionary, developmental, and functional mechanisms; a different environment will result in different structure and function at each level.

**Complex systems: Biology:** Emergent laws arise out of these foundations, through coarse-graining, symmetry breaking (Anderson 1972 [15]), or black boxing (Marshall et al., 2018 [16]). As regards to biology (Campbell and Reece 2005 [17]), there are well-established effective laws for: Physical chemistry (Atkins and de Paula 2011 [18]);Macromolecular chemistry (Lehn 2004 [8]);Molecular biology (Watson et al., 2014 [9]);Cell biology (Allen and Cowling 2011, Alberts et al., 2015 [19,20]), including cell signaling networks (Berridge 2014 [21]), metabolic networks (Jeong et al., 2000 [22]), and gene regulatory networks (Karlebach and Shamir 2008 [23]);Physiology (Hall and Hall 2020 [24], Davies 2021 [25]);Neuroscience (Kandel 1991 [26], Churchland and Sejnowski 1994 [27], Kandel et al., 2000 [28], Kandel 2012 [29]);The psychological level in some instances (Franklin et al., 2013, 2016 [30,31], Jacob et al., 2023 [32]).

These are often called “special sciences”. Putting them together, one arrives at the hierarchy of emergence depicted in Figure 1 below. There are also emergent laws for origins of these entities, firstly as regards evolution, occurring on long timescales, with some species dying out and new species coming into being through the processes of natural selection (Darwin 2019 [33], Campbell 1974 [34], Dobzhansky 2013 [35], Wagner 2014 [11]). Development occurs on shorter timescales (Gilbert 2001 [36], Wolpert et al., 2002 [37], Gilbert 2019 [38]). These interact in an EVO–DEVO way (Gilbert et al., 1996 [39], Oyama et al., 2001 [40], Carroll 2005, 2008 [41,42]). These processes operate simultaneously as regards all the aspects 1.–6. enumerated above (Ellis 2023 [43]: §3.2). A key feature is that information flows play an important role in biological function (Nurse 2008 [44], Farnsworth et al., 2013 [45], Farnsworth 2022 [46]).

**Complex systems: Engineering, computers, the internet:** Complex systems created by human beings arise on the same foundation of particles and atoms, designed on the basis of the same principles of modular hierarchical structuring (Booch et al., 2007 [12]). This provides the foundations of technology (Arthur 2009 [2]) and engineering (Blockley 2012 [47]) based in the principles of engineering design (Dieter and Schmidt 2021 [48]).

**Figure 1 entropy-25-01301-f001:**
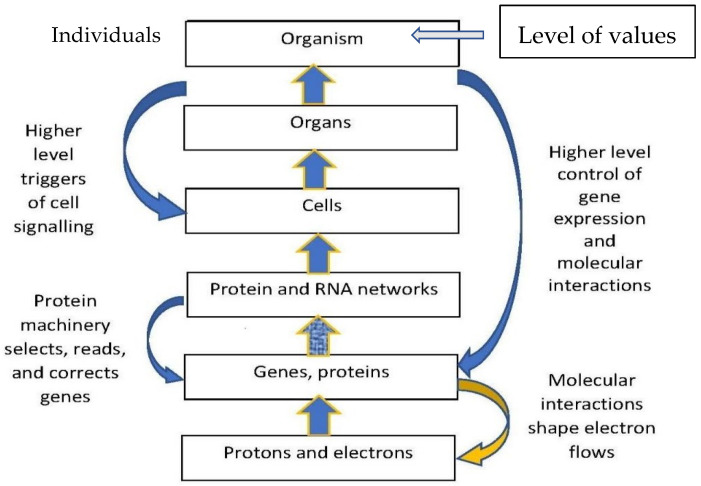
**The interplay between bottom-up and top-down interactions in an individual** (Ellis and Noble 2023 [49]). Causal closure occurs because the upwards arrows range from the bottom physics level to the level of the whole individual, and the downward arrows reach all the way down again, linking all levels (Ellis 2020a [50]). Values, related to final causation, occur at the level of the individual person. There is a higher level (“Society”) not shown here.

### 1.3. Updated Versions of Aristotle’s Causes

The central theme of this paper is that updating Aristotle’s four causes to take into account these more recent discoveries will form an important resource for understanding complex emergence these cases and that doing so validates the emergentist viewpoint: each of the emergent levels in the hierarchy of existence has real causal powers (Clayton and Davies 2006 [51], Murphy et al., 2009 [52], Noble 2012, 2016 [3,53], Ellis 2020a [50], O’Connor 2021 [54]).

Previous references that have looked at this issue include Ackoff (1973) [55], Juarrero (2002) [56], Noble 2016 [53], Ellis 2012 [57], Tabaczek 2013 [58], Ellis 2016 [59], Hofmeyr 2017, 2018 [60,61], and Noble and Noble 2021 [62]. This paper develops, in depth, the way generalized versions of Aristotle’s four causes enable emergence in biology and technology, by showing in detail how formal and material causes enable downwards causation, which together with upwards emergence underlies the way efficient causation occurs at every emergent level in the hierarchy (Noble 2012 [3], Ellis 2020a [50]) and hence allows final causation as well. 

**Causation:** What notion of causation do we use in the following discussion? It will be defined in terms of ***difference making***, whether experimental or counterfactual (Menzies and List 2010 [63], Kment 2010 [64], Pietsch 2016 [65]), but noting that determining what is in fact making a difference may require in depth investigation (Pearl and Mackenzie 2018 [66]). One way to achieve better assessments of causation is to construct multi-factorial models, from which the causal role of each factor can then be unraveled through defining all the conditionals under which the causal role of any particular factor can be deduced (I thank Denis Noble for this comment). This also assumes that time passes with a unique direction of time, which is indeed the case on our current cosmological view (Ellis 2022 [67]), despite some counter claims based in the idea of a block universe. Thus, there is no ambiguity as regards to causes preceding effects.

### 1.4. This Paper

The sections that follow are: Section 2—The hierarchy of existence; Section 3—Efficient causation; Section 4—Formal causation; Section 5—Material causation; Section 6—Final causation; Section 7—The outcome: the significance of Aristotelian causation, updated.

## 2. The Hierarchy of Existence

Putting together the scientific understandings just discussed (Section 1.3), we understand emergent systems as being adaptive modular hierarchical structures. These form the context for the functioning of the four causes. I discuss now modular hierarchical structures (Section 2.1), the issue of reductionism and emergence in this context (Section 2.2), and the relation to the four causes (Section 2.3).

### 2.1. A Current View: Modular Hierarchical Structures

**Biology**—Structure enables function (Brewer and Burrow 1980 [68]), which is central to biology (Hartwell et al., 1999 [69]) (See *Human Biology*, Chapter 10: Structure Determines Function (Salt Lake City College) for many examples.). If we conjoin the effective laws **1–7** discussed in Section 1.3, we get the hierarchical structure represented in Figure 1. It is modular, with each level being made up of semi-autonomous components interacting with each other.

**Why modular?**—The basic principle underlying complex emergence is to break up a complex task at level **L** into simpler tasks, create modules at level **L-1** to handle the simpler tasks, then combine the outcomes of the modules to handle the higher-level task. The modules at level **L-1** in turn can be broken up into sub-modules to handle even simpler tasks, till a level **L0** is reached, where the needed action is so simple it can be fully handled at that level, in general, the interaction at that level will be linear (Ellis 2023 [43]). 

**Why hierarchical**?—Hierarchy naturally emerges on carrying out an integration of the modules. Vast numbers of modular components are involved at lower levels. The number of cells in a human body is 10^13^, number of proteins in a cell is 10^7^, number of atoms in a cell is 10^14^, number of atoms in a human is 10^27^. The lower levels have a highly dynamic nature: molecules in air at normal room temperature are moving at between 300 to 400 m per second. A biomolecule collides 10^13^ times a second with water molecules at room temperatures (see “*Cells are very fast and crowded places*” by Ken Shirriff), as is evidenced by Brownian Motion (Hoffmann 2012 [70]). Macromolecules change shape as they carry out biological functions. 

**Nature of the modules:** Key features of modules are characterized by (Booch et al., 2007 [12]) as follows:**Boundaries and binding:** they have some kind of boundary limiting their extent and are more tightly bound with faster internal interactions than the weaker strength and slower speed of interaction between modules.**Information hiding**: a user of a module does not need to know the internal variables and mechanisms of the module; all they are concerned with is that it carries out its required tasks.**Abstraction** (**black boxing**): the functioning of the module can be characterized by an abstract description relating the input variables to output variables (Ashby 2013 [71]), summarized by a suitable label/name.**Interfaces:** control transfer of energy and information in and out of a module according to a protocol.**Multiple realizability:** A key feature is the multiple realizability of a module’s internal structure consistent with its abstract description and interface protocols (Gillett 2002 [72], Piccinini and Maley 2014 [73], Batterman 2018 [74], Bickle 2020 [75]). This is an inevitable result of evolutionary origins (Edelman and Gally 2001 [76]).**Origin**: Modularity makes possible the coming into being of very complex structures through the inheritance of modules with variation and selection; otherwise, it is simply not practicable (Simon 2019 [77]).

**Function:** The definition of function in biology has been subject to some debate. We use an ***organizational account*** of biological functions (Mossio et al., 2009 [78], Farnsworth et al., 2017 [79], Ellis and Kopel 2019 [80]); in essence, functions are there in order to enable the organism as a whole to function on a day-to-day basis and the whole is organized to make this possible. Thus, this definition does not depend on evolutionary origins (see the discussion in Gardner 2009 [81].

**Technology:** The same principles apply to technology (Arthur 2009 [2], pp. 32–43), including digital computers (Tanenbaum and Austin 2013 [82], Ellis 2016: §2 [59]) and materials (Fratzl and Weinkamer 2007 [83], Chen et al., 2021 [84]). In these cases, we might refer to *structure and dynamics* (Dove 2013 [85]) rather than *structure and function* (Brewer and Burrow 1980 [68]), although in all cases it might be appropriate to still use the term “function”, e.g., in the case of manufactured materials (Miodownik 2014 [86]) and in the case of digital computers (Ellis and Drossel 2019 [87]), as they all serve a purpose. In each case, there is a hierarchy of emergence similar to that shown in Figure 1 (Batterman 2013 [88], Ellis 2016 [59], Ellis and Drossel 2019 [87]).

### 2.2. Reductionism and Holism

The emergent whole arises out of the parts: it could not exist without them. Given the context of the emergent hierarchy (Figure 1), the issue arises of how the emergence of the higher levels from the lower levels occurs, both diachronically (over time) and synchronically (at each time). This occurs diachronically by assembly in *chemical processes* (Atkins and de Paula 2011 [18]), *developmental processes* (Gilbert 2019 [38]), or *manufacturing* (Kaeslin 2008 [89]), each case involving symmetry breaking (Anderson 1972 [15]). It occurs synchronically by *coarse-graining* (Flack 2017 [90]) or *Black boxing* (Ashby 2013 [71], Marshall et al., 2018 [16]) via logical combinations of variables, as in digital computers (Booch et al., 2007 [12], Dasgupta 2016 [91]).

There are two essentially opposing views about the resultant emergent levels that have been strongly debated for many decades: 

**A reductionist view**: The whole is *nothing but the sum of the parts*, even if we cannot actually predict outcomes at macrolevels from details at microlevels. It is claimed by some physicists that, in reality, nothing takes place except interactions between particles (Greene 2020 [92], Carroll 2021 [93], Hossenfelder 2022 [94]). No essential dynamics takes place at any higher levels. 

**An emergentist or holistic view**: Irreducible emergent entities arise that are *more than the sum of the parts*: summation of lower-level effects does not suffice to explain emergence (Clayton and Davies 2006 [51], O’Connor 2021 [54]), with various kinds of emergence proposed (Cunningham 2001 [95], Chalmers 2006 [96], Brigandt and Love 2023 [97]). Paoletti (2020) [98] characterizes emergence as partial and qualified dependence of the emergent entities on the entities in their emergence bases. Santos (2021) [99] characterizes it in terms of a relation of both partial dependence and partial independence (the latter being conceived in terms of a relative irreducibility or autonomy) between a given entity (the putative “emergent”) and the entities with respect to which it may be said to be emergent (the “emergence base”). This emergence is enabled by downward causation occurring in addition to bottom-up causation (Noble 2012 [3], Ellis 2012 [57]). The summary slogan that captures the emergentist theme is: 


*“The whole is greater than the sum of its parts.”*


This is a common claim nowadays amongst those studying complex systems, see for example Ackoff (1973) [55], Tanaka et al. (1998) [100], Glaser and Glaser (2000) [101], Galatzer-Levy (2002) [102], Davison et al. (2006) [103], Pérez-Dorado et al. (2012) [104], Matthews and Guarné (2013) [105], Kornblith et al. (2014) [106], Taylor and Wood (2019) [107], and Fang and Hu (2022) [108]. However, it is denied not just by some reductionist physicists, as mentioned above, but also by reductionist philosophers (Kim 2005 [109]), biologists (Dawkins 2016 [110]), and neuroscientists (Crick 1994 [111]), denying the possibility of free will (Harris 2012 [112], Caruso 2012 [113]). 

We note here that there is a major problem with these claims: ***they cannot all be right***! If physics unequivocally causes all, then neither genes nor neurons have any real causal powers; whereas if either of these emergent entities have real causal powers, then it is not true that physics by itself determines all. These claims contradict each other. 

Which of them, if any, is correct? The solution is that *all emergent levels have real causal powers* (Noble 2012, 2016 [3,53], Ellis 2020a [50], Noble and Ellis 2022 [114]), as discussed in §3. Consequently, these levels exist ontologically, not just epistemologically: we reliably determine this ontological nature by suitable experiments. This is supported by Simpson and Horsely (2022) [115], who state:


*“We argue that the causal powers of many systems are determined by) higher-level, macroscopic properties that are neither reducible nor weakly emergent, and that contemporary physics is compatible with some kind of pluralism that affirms that these entities are robustly real.*
*”*


Again, I note that I am not concerned here with the debate about the difference between weak and strong concepts of emergence, that is, the question of whether one can uniquely determine higher-level outcomes on the basis of lower-level dynamics and variables alone. Rather, the claim is that verifiable emergent-level dynamics does indeed occur at higher levels, causally affecting outcomes, and indeed this is why such emergent levels are recognized as existing and being characterized by “special sciences”—which indisputably exist as academic disciplines. Their basis in lower-level dynamics is indeed true, but that lower-level dynamics is influenced in a downward way through mechanisms I discuss is how interlevel relations are coordinated and leads to causal closure. 

### 2.3. Relation to the Four Kinds of Causes

Given the context of the hierarchy of emergence (Figure 1) that I have just sketched, with the associated effective laws (Section 1.3), we can now state the main theses that will be developed in what follows. They relate to the hierarchy of emergence as follows: ***efficient causation*** occurs at every emergent level in biology and technology (Noble 2012 [3], Ellis 2020a [50], Ellis and Noble 2023 [49]). Although it is enabled by both formal and material causation, they can each be distinguished from the other. This same level causation is enabled by the ***interplay of upwards and downwards causation*** between emergent levels. Upwards causation takes place by ***coarse-graining***, including ***structural emergence*** and ***black boxing***. The two key forms of downward causation enabling this are ***formal causation*** and ***material causation***, which occur in both synchronic and diachronic forms. ***Final causation*** is a further form of downward causation in the cases of intelligent beings (Noble and Ellis 2022 [114]) and so for the social structures and artefacts they create (Ellis and Drossel 2019 [87], Ellis and Noble 2023 [49]). ***Causal closure*** only occurs when all inter-level linkages generated by all these forms of causation are taken into account (Montevil and Mossio 2015 [116], Ellis 2020 [117]). No physical level is, by itself, causally complete, as claimed by some (Kim 2005 [109], Hossenfelder 2022 [94]). Such a claim ignores all the factors that lead to specific physical outcomes at a particular time and place. Physical laws per se, expressed for example by Newton’s laws of motion and by Maxwell’s equations, do not lead to any specific outcomes at all.

### 2.4. Is This Persuasive?

I have received the following comment from a referee:


*“But there is little work done to actually prove, to a skeptical reader, that this is indeed the right way to think about things. The rejoinder could always be that such causal descriptions are merely convenient and that all the real work is still being done at the level of physical forces—it’s just too complicated for us to grasp”.*


My response is that whereas real work is certainly being conducted at the level of physical forces, “real work” is also being conducted at each emergent level, as is shown by all the examples given. It is, for example, an undoubted fact that genes can be read by cell machinery in such a way as to produce proteins, following well-established causal relations, and that the pumping of blood by the heart keeps cells alive, as proven by physiological studies. These are factual statements proven over and over again by experiments and indeed by biological functioning in daily life. They are independent of any views on reduction and emergence. The idea that any given level has some causal independence from the goings-on at lower levels relies on ideas of hierarchy, modularity, coarse-graining, multiple realizability, and causal protectorates, which I argue in this paper. Yates (2017) [118] contends that examples such as those I give demonstrate the reality of strong emergence. O’Connor (2021) [54] comments: 


*“A skeptic might press that the effects Yates cites as pointing to a distinctive kind of higher-level causal power are themselves all higher-level. Assuming that all macroscopic properties are microphysically realized, if one were able to take a wide-angle view of the evolving process in purely micro-physical terms throughout (including in characterizing the targeted token effects), it’s not clear that reference to anything other than the features of and basic relations among microphysical entities is required for explanation. It might well be the case that to explain the token effects under their macroscopic description requires equally macroscopic appeal to molecular geometry (where a given geometric shape is multiple realizable by distinct spatial arrays of atoms). But such explanatory irreducibility is, as we’ve seen, the hallmark of forms of weak emergence.”*


However, as stated above, I am not here concerned with the distinction between strong and weak emergence but rather that the higher levels are indeed examples of emergence whether strong or weak, and that is confirmed by giving examples where it is indeed true. In any case, an Aristotelian approach to emergence is helpful at this point, since the Aristotelian can insist that we cannot fully describe the microscopic level except in terms of the specific ways in which that level is informed and structured by higher levels.

## 3. Efficient Causation

This section looks at how an updated version of efficient causation works out in the context of our present-day understandings of science. It considers the nature of an updated version of efficient causation (Section 3.1) and how this works out in biology (Section 3.2) and technology and digital computers (Section 3.3). It finally considers abstract causation (Section 3.4).

### 3.1. Updated Version of Efficient Causation

Aristotle’s concept of efficient causation was presented in Section 1.2. He was not in a position to differentiate the different levels of emergence indicated in Figure 1, as the relevant scientific results leading to that hierarchy were not then known. Given this hierarchy, we follow (Noble 2012 [3]), Ellis (2020a) [50], and (Ellis and Noble 2023 [49]) by proposing the following updated version. 

Each emergent level **L** for a system **S** is characterized by suitable variables **X^i^_L_**(**t**) at time **t** (Anderson 1972 [15]). In this context, our updated definition of material causation is:

***Efficient causation*** *takes place at an emergent level **L** in the hierarchy of emergence of a system **S** when effective laws **E_L_**(**X^i^_L_**(**t**)) at level **L** determine the values at times **t > t_0_** of the relevant variables **X^i^_L_**(**t**) at that level from initial data **X^i^_L_**(**t_0_**) at time **t_0_**. This is a characterization of causation at that level.*

The effective laws may be deterministic, as in Newton’s laws of motion and Maxwell’s equations, or probabilistic, as in the case of statistical mechanics and quantum physics; they may even involve chaotic dynamics (Shen et al., 2021 [119], Neyrinck et al., 2022 [120]). Some key points:Emergent levels are *defined* by the existence of efficient causation (i.e., well-established effective theories) and associated variables at that level. It will generally take place via a combination of formal and material causation at that level.Efficient causation at an emergent level **L** is enabled by a combination of upward and downwards causation between levels, the latter occurring via formal and material causation, as we discuss below.All these effective levels and associated laws are needed in order that the system functions: if any of them are missing, it will not work. This is Noble’s *principle of biological relativity* (Noble 2012, 2022 [3,121]), which can be extended to emergent contexts of technology (Ellis 2020a [50]).There can be no efficient physical causes without some non-random organization of stuff—a directionality of physical forces arises from physical inhomogeneities. Thus, the notion is intimately linked with the idea of symmetry breaking (Anderson 1972 [15]).

Some physicists and philosophers only recognize efficient causation at an underlying physics level without prescribing what that level is (they face the embarrassment that we do not know what the bottom-most physics level is). For practical purposes in biology and engineering and indeed everyday life, the relevant underlying effective physics theory is the level of the time-dependent Schrodinger equation for electrons and nuclei, set out in Laughlin and Pines (2000) [122]. However, that is emergent from lower physical levels: it is based in a lower level, the level of the standard model of particle physics (Oerter 2006 [123]) (described in Wikipedia here: https://en.wikipedia.org/wiki/Standard_Model (accessed on 11 June 2023)), itself believed to be based in some (as yet unknown) lower level, that of a “Theory of Everything” (perhaps string theory/M theory, perhaps not). In contrast, we claim that efficient causation takes place and is causally effective at every emergent level L.

### 3.2. Biology

Broad categories of such emergent laws were given in Examples **2–6** listed in Section 1.3. Some specific cases with well-defined effective theories are cell signaling processes (Berridge 2014 [21]), gene regulatory networks (Davidson and Erwin 2006 [33], Karlebach and Shamir 2008 [23]), metabolic networks (Jeong et al., 2000 [22]), the heart (Noble 2002 [124], Fink and Noble 2008 [125]), nerve cells and behavior (Kandel 1991, 2001 [26,126]), and neural networks (Churchland and Sejnowski 1994 [27], Carpenter and Grossberg 1987, 1988 [127,128]).

### 3.3. Digital Computers and Technology

Digital computers are designed according to the precepts of modular hierarchical structures (Simon 2019 [77], Booch et al., 2007 [12]). The emergent levels are characterized in Tanenbaum and Austin (2013) [82]: there is a tower of virtual machines, linked by compilers or interpreters (Ellis 2016: §2, [59]). Algorithms drive what happens (Dasgupta 2016 [91], Ellis and Drossel 2019 [87]), and they too are hierarchically structured in a modular way. 

All engineering (Blockley 2012 [47]) is similarly based in modular hierarchical structures, for very good reasons (Simon 2019 [77]). For example, an aircraft is a hugely complex structure (see for an indication of this hierarchy https://mae.ufl.edu/haftka/structures/FAA.pdf (accessed on 11 June 2023)), including many computers. Each form of engineering is based in some technology (Bronowski 2011 [129], Arthur 2009 [2]), which is the effective causation enabling that engineering practice. Thus, there is *civil engineering*, enabling stable structures (bridges, roads, tunnels, buildings, dams) to exist, based in the design and construction of structures so as to ensure their stability; *mechanical engineering*, based in the control of energy and particularly thermodynamics; *electrical engineering,* based in the control of the flow of electrons and associated magnetic fields; *electronic engineering*, based in logical operations enabled by transistors and integrated circuits; *chemical engineering*, based in the control of chemical interactions and purification processes; *biotechnology*, based in molecular biology interactions and particularly now the use of CRISP-R; *nuclear engineering*, based in nuclear interactions; *information technology*, based in digital computers and nowadays engaging in AI projects. The latter is more than computer science: it refers to how to gather, analyze, sort, and use information productively. 

In each case, efficient causation takes place at multiple levels in an integrated way involving both material and formal causation at each level, the higher levels being designed to shape the lower-level dynamics so as to produce the desired higher-level outcome (Dieter and Schmidt 2021 [48]). The emergent results (e.g., an aircraft flies with specific performance capabilities) are determined and verified as effective laws at the integral emergent level, which can be determined without any knowledge of the lower-level structures and dynamics (a pilot does not have to be a quantum physicist). All these examples are clear cases of the whole being greater than the sum of the parts: it is the specific organization of the parts that leads to the desired outcomes. Each of these technologies has emerged through an evolutionary process (Arthur 2009 [2]), exploring the possibilities within the boundaries allowed by physical constraints (Vogel 2000 [130]). 

### 3.4. Abstract Causation

Because of our symbolic abilities (Deacon 1998 [131]), in the case of human beings, abstract causation takes place: that is, ***abstract entities have causal powers***. This occurs through the interactions between individuals and the society in which they live, for it is through that interaction that causal powers of abstract entities arise. They do not arise because of any individual’s brain state, because it is *socially shared concepts* that have the causal power (Murphy and Brown 2007 [132]). 

An example is the ***rules of chess***, which are abstract concepts stated as formal rules. These do not derive their causal power though being instantiated in any one specific person’s brain: these powers arise from them being instantiated in the brains of every person who plays chess. Furthermore, they are not only instantiated in neural states in brains: they are instantiated in chess books, in algorithms in computer programs, talks given on the internet, in the moves on a chess board, and so on. Conceptually, the rules of chess are the abstract *equivalence class of all such instantiations* (Noble and Ellis 2022 [114], Ellis and Noble 2023 [49]). 

Other examples are, *plans* enabling the creation of artifacts such as buildings, bridges, aircraft, computers (Ellis 2016 [59]); *money*, used as a medium of exchange or a store of value (Davidson 1972 [133]); *signs*, such as those indicating restrooms for males and females; *laws*, such as those for motor vehicle behavior at traffic intersections, instantiated by stop signs and by traffic lights that cause drivers to stop when red and allow them to move when green; and *legal agreements,* underlying the existence of organizations such as closed corporations (Harari 2014 [134]). In each case, the entity is abstract because it is multiply realizable, for example plans for an aircraft can be talked about, drawn on paper, instantiated in computer programs, and listed in a formal specification; the abstract concept is not the same as any specific one of these instantiations. Each of them in turn have causal powers through the brain functioning underlying mental causation (Menzies 2003 [135], Robb and Heil 2021 [136]) and taking place in a social context (Aberle et al., 1950 [137]) that determines the appropriate functioning of these abstract entities. 

The key point them is that ***these can all be regarded as forms of efficient causation taking place at the level of the individual person in Figure 1***, occurring in the current social context that both enables it to occur and provides constraints on what kinds of abstract causation are allowed.

## 4. Formal Causation

Aristotle’s concept of formal causation was presented in Section 1.2. This section looks at how an updated version of formal causation works in the context of our present-day understandings of science. It considers an updated version of formal causation (Section 4.1), formal causation and life (Section 4.2), in artefacts and engineering (Section 4.3), oscillators (Section 4.4), feedback control/homeostasis (Section 4.5), and adaptive brain networks and ion channels (Section 4.6).

### 4.1. Updated Version of Formal Causation

Structure, which can be labelled “form”, underlies function at every emergent level in biology (Brewer and Burrow 1980 [68], Wainwright 1998 [138]) and engineering (Blockley 2012 [47]). In agreement with Aristotle’s understanding, this can be suitably labelled “formal causation”. It takes place either synchronically (the form is unchanging and shapes interactions in a fixed way) or diachronically (with form responding dynamically to changing conditions). Form has causal powers by imposing constraints at higher levels which then entrain lower-level variables. Crucially, it can occur at any single level **L**, then being a mechanism whereby efficient causation takes place at that level by setting constraints at that level, but also acts down to shape what happens at lower levels **L − N** by setting symmetry breaking constraints at those levels. Because of this possibility, efficient causation at level **L** can be shaped by formal causation at higher levels **L + M**. Our updated definition of formal causation in the context of the relevant hierarchy of emergence is: 

***Formal causation*** *takes place at an emergent level **L** in the hierarchy of emergence of a system **S** when the variables **X^i^_L_**(**t**) in the effective laws **E_L_**(**X^i^_L_**(**t**)) at level **L** of **S** obey constraints **C**(**X^i^_L_**(**t**)**,t**) **= 0**. The constraints may depend on time via higher level variables **X^i^_M_**(**t**)**, M > L**: then **C**(**X^i^_L_**(**t**)**,t**) **= F**(**X^i^_L_**(**t**)**, X^i^_M_**(**t**)) **= 0**. Such constraints determine the shapes of entities and perhaps how that shape changes with time.*

Indeed, *constraints are causes* (Juarrero 2002 [56]), given our definition of causality. There is a paradox here: without constraints, one just has particles randomly bumping into each other. Adding constraints can generate functionality (Deacon 2006 [139]); for example, an electric wire channels electrons from point A to point B along the wire, the point being that the electrical resistance orthogonal to the direction of the wire is essentially infinite because of its insulating sheathing. Thus, its function is to channel the flow of electrons at the electron level in order to fulfil some function at a higher level. Because of this constraint, the electrons do not diffuse randomly in all directions, as occurs in Brownian motion.

Because it’s effect chains down from the level **L**, *constraints at level **L** affect all lower levels **N < L***. Thus, formal causation at level L exerts a form of ***downward causation*** (Noble 2008, 2012 [3,140]). There are a great many detailed ways constraints at a higher level can be realized at lower levels. Thus, as is usual in downward causation, one has ***multiple realizability*** of higher-level constraints at lower levels.

Static (synchronic) or dynamic (diachronic) formal causation: if ∂*C(X^i^_L_(t),t)/∂t = 0* in the interval I*: t*_1_ *< t < t*_2_, then *formal causation is static* during I, remaining unchanged during I; if ∂*C(X^i^_L_(t),t)/∂t ≠ 0 in I*, then *it is dynamic* during I. 

When the constraints depend on time via higher level variables, dynamic formal causation enables ***contextual branching of dynamics to meet higher level needs***. As an example, consider electric wiring joining one terminal of a battery to a switch, then a light, and then the other terminal of the battery. The following branching dynamics arise at the macrolevel from the constraints created by the circuit:IF {switch is on} THEN {light shines} ELSE {not}. (1) The resulting macro- to microlevel relation is:IF {switch is on} THEN {electrons flow in wire and lamp} ELSE {not},(2)
whereas the companion micro- to macrolevel relation is:IF {electrons flow in lamp} THEN {lamp shines} ELSE {not}. (3) Thus, macrolevel constraints control microlevel dynamics, which then cause macrolevel events through the closed form of the circuit at the macrolevel when the switch is closed but its open form if the switch is open (a macrolevel difference in circuit topology that cannot be described at the microlevel). The higher-level need being met is to attain the right level of light in the room. This also demonstrates how the human mind/brain can downwardly control the flow of electrons in a wire by changing the switch setting, which happens every time someone turns a light on or off. 

Is formal causation always associated with function? Not necessarily; there are a multitude of natural forms out there—planets, rocks, mountains, rivers, lakes, oceans—that can have complex forms and associated formal causation without associated functions. However, all living systems are associated with functions (Hartwell et al., 1999) [69], as are all artefacts (Simon 2019 [77]), engineering systems (Blockley 2012 [47]), organizations (Etzioni 1964 [141]), and social structures (Elder-Vass 2010 [142]), so formal causation in all these contexts will be associated with function. In all the latter cases, this occurs via agency exerted by the human mind in a social context (Murphy and Brown 2007 [132], Noble and Noble 2021 [62]).

### 4.2. Formal Causation and Life

Our bodies have an immensely complex emergent form, characterized by our physiology (Hall and Hall 2020 [24], Davies 2021 [25]), that enables our functioning. This occurs via time-dependent formal causation enabled by that very complex modular hierarchical structuring. Crucially, the link between physics and biology is macromolecular chemistry (Lehn 2004 [8]), where it is the conformational change of macromolecules—a change of shape—together with the lock and key mechanism of molecular recognition (Behr 2008 [143]) that is the key principle of molecular biology in general and catalysis by enzymes in particular. 

### 4.3. Formal Causation: Artefacts and Engineering

A vase or jug contains water because of its “container” shape; it would not do so if that shape were different because of a crack or hole in it. The shape of the vase causes the water to be contained. A pipe leads water from one place to another; its form prevents the water from spreading out sideways and takes it to its appointed destination. Similar examples on a larger scale are a tunnel, a bridge, an aircraft landing strip, and a dam, each designed with function in mind. The shape of a piano or cello is carefully designed to produce specific kinds of musical sounds through its shape (and its constituent material: material causation also takes place). 

A building is another example, providing shelter and functional spaces of various kinds (cooking, sleeping, relaxing, bathing, and so on). In this case, *dynamic* formal causation can take place, as doors may open and close, allowing ingress and egress or not, curtains may be open or drawn, allowing light in from outside or not, and so on. Thus, the nature of the constraints resulting in formal causation can be adjusted to meet needs that vary with time. 

Each form of engineering (Blockley 2012 [47]) is based in formal causation associated with its underlying technology (Arthur 2009 [2]) and has been designed to be that way through the general principles of engineering design enabling the desired outcomes (Dieter and Schmidt 2021 [48]) (for a great discussion of a specific case, see the aircraft design process on Wikipedia). Thus, electrical engineering is based in wires, resistors, capacitors, electromagnetic coils, magnets switches, relays, light bulbs and LEDs, sensors, batteries, electric motors, and dynamos. The specific configurations of these components is the formal cause of outcomes such as for automobile ignition systems, lighting systems, functioning of domestic appliances, and so on. Electronic engineering, based additionally in transistors and integrated circuits and further components such as microphones, loudspeakers, and digital cameras based in charge-coupled devices (CCDs), allows the construction of music systems, digital computers, electronic alarm systems, automatic pilots, etc.

In both cases, one has *dynamical formal causation* allowing very complex conditional branching of dynamics of modular hierarchical structures to produce the desired functionality. A key issue here is how to design a suitable user interface for such systems, which is a complex topic in its own right (Norman 1986, 2013 [144,145]) relating formal to final causation.

### 4.4. Formal Causation: Oscillators

Oscillators are a key emergent feature of engineering, biology, brain function, and social life, where they underlie clocks. They are all outcomes of formal causation of varied kinds, occurring because of the shapes of emergent structures. 

In engineering, examples are that steam engines, petrol engines, and diesel engines are all powered by recurrent mechanical cycles generated by the shapes of the mechanical components and their interlocking structure. Electronic oscillators are widely used, for example in radios, inverters, digital games, and computers. The functioning of digital computers is centrally based on an IPO cycle: input → process → output, generated by an electronic oscillator. It has a system clock with various functions. 

The principles of biological oscillators are set out in Novak and Tyson (2008) [146] and Li and Yang (2018) [147]. They occur in cells, where cell-cycle clocks and switches underlie cell development processes (Tyson and Novak 2022 [148]); the heart (Noble 2002 [124], Fink and Noble 2008 [125]) and lungs, where each have regular cycles that jointly keep us alive by providing our cells with the oxygen and nutrients they need; and nervous system. The stomatogastric ganglion of the spiny lobster (Churchland and Sejnowski 1994 [27]) controls gastric mill muscles that open and close a set of three teeth in a kind of chewing behavior. This repetitive behavior is generated by small neural circuits that essentially forms an oscillator (Daur et al., 2016 [149]). Metastable oscillatory modes emerge from synchronization in the brain spacetime connectome (Cabral et al. (2022) [150], and adaptive resonant networks occur in the brain (Carpenter and Grossberg 1987, 1988 [127,128]). 

### 4.5. Formal Causation: Feedback Control/Homeostasis

A crucial feature of engineering and biology is feedback control, labelled *cybernetics* in an engineering context (Wiener 1961 [151], Ashby 1964 [152]) and *homeostasis* in a biological context (Modell et al., 2015 [153], Hall and Hall 2020 [24], Ellis and Nobel 2023 [49]). The problem is to maintain stability or attain a desired goal in the face of ongoing perturbing influences. The solution is to make continual adjustments to counteract the effect of such disturbances using information flows to attain the desired outcomes. 

**The Basic Principle**: Suppose a system **S** is in a state **SS**, the desired state is **DS**, and **SS** is measured by a sensor **M** to be different from **DS**, so there is an error **ES:= DS − SS** between the desired and actual state. This error is sent by a controller **C** via a physical connection to an activator **A** that can alter the system state **S** in such a way as to reduce the magnitude **|ES|** of the error **ES**. This is performed until the error **ES** is less than some threshold **ΔS** below which the difference does not matter. Then, the activator is turned off. 

**Example**: A thermostat for a room has a controller **C** where you set the desired temperature **DT** and a sensor **M** that determines **T**. The controller can turn on a heater **A**, which will increase the temperature **T** until |**DT − T| < ΔT**. The **branching logic** that emerges is:IF {|**DT − T| > ΔT}** THEN {**C** activates **A} ELSE** {**A** is turned off}. (4) This is formal causation because the logic (4) emerges from the specific form of the system: the emergent macrolevel loop that connects the components **M**, **C**, and **A**. If the form is changed by disconnecting the activator, then the outcome is different: all the parts are still there but the topology of the circuit is open, not closed; the circuit form is changed, and no homeostatic action takes place. It is downward causation because the system acts to attain the desired temperature **DT** at the macrolevel, thereby controlling the average molecular speed **<v^2^>** at the microlevel downwardly, because **<v^2^> = α T** where **α = 3R/M**.

Homeostasis is a central principle in control engineering, organizations (Penrose 1952 [154], Scott and Davis 2007: pp. 90–93 [155], Damasio and Damasio 2016 [156]), and biology at every level. We have homeostatic systems for pH levels, blood glucose, iron levels, levels of blood gases, calcium and sodium concentrations, and cell membrane potentials at the microlevel and body temperature, blood pressure, normal heart rate, and fluid balance at the macrolevel; you are ill if any of these quantities are out of bounds, which is why these levels are used for medical diagnoses of health. 

### 4.6. Formal Causation: Adaptive Brain Networks and Ion Channels

The central nervous system (CNS) has several layers of structure (Churchland and Sejnowski 1994: Figure 1.4 [27]), namely, molecules, synapses, neurons, networks, maps, systems, and the CNS as a whole. Systems have topographic maps, layers, and columns and contain local networks. Neurons have a complex structure of axons, a nucleus, and dendrites, joined to other neurons by synapses, so forming networks (Churchland and Sejnowski 1994: pp. 27–60 [27]). These networks are shaped by our life experiences through various mechanisms of brain plasticity, resulting in the unique total set of synaptic connections that make each of us the individual that we are: this is the dynamic form of our individual brain. The connections synaptic strengths are continually changing as we interact with the physical and social world around us. This is the time-dependent formal cause of all the brain dynamics underlying our thought and actions (Leopold 2023 [157]).

At the neuronal level, action potential spike chains convey signals from one neuron to another, enabled at the molecular level by flow of sodium and potassium ions through the cell membrane, controlled by voltage-gated ion channels. These are proteins imbedded in the cell membrane that change conformation in response to the voltage across the cell membrane in according to the way supramolecular chemistry works by conformational changes (Lehn 2004 [8]). These changes reach down to affect the level of electrons by changing constraints in the underlying Hamiltonian (Ellis and Kopel 2019 [80]), thus enabling contextual branching logic to emerge from the underlying physics at the neural level and hence at the emergent network levels (Kandel et al., 2000 [28]).

## 5. Material Causation

Aristotle’s concept of material causation was presented in Section 1.2. This section considers an updated version of material causation (Section 5.1) and then its application at the basis of engineering, technology, and daily life (Section 5.2) and how it occurs in metabolism and the cardiovascular system (Section 5.3), in gene regulatory networks and developmental processes (Section 5.4), and in adaptive selection and evolutionary processes (Section 5.5).

### 5.1. Updated Version of Material Causation

Material causation either takes place synchronically (statically): materials are there at the present time due to previous processes, or diachronically (dynamically): material stuff is continually being altered to fit same-level or higher-level needs, their existence and nature is due to ongoing dynamic processes. This dynamic aspect is key to biology. Either can occur at level **L**: efficient causation takes place at that level through the nature of the material stuff present at that level. In the dynamic case, it can also act down to shape the nature of material stuff at lower levels **L − N**. Because of this possibility, conversely, efficient causation at level **L** can be shaped by material causation at higher levels **L + M**. 

The proposed updated definition of material causation in the context of the relevant hierarchy of emergence is: 

***Material causation*** *takes place at an emergent level **L** in the hierarchy of emergence of a system **S** when a set of variables **{X^i^_L_**(**t**)**}** characterizes the nature of material stuff at level **L** of **S**. This underlies the possible emergent set of such variables **{X^i^_N_**(**t**)**}** at levels **N > L** of **S**. Material causation is **static** in an interval **I = [t1, t2]** if the set of variables **{X^i^_L_**(**t**)**}** is constant in that interval and is **dynamic** if **{X^i^_L_**(**t**)**}** changes during this time.*


*It affects higher levels **L > M** upwardly and can affect **lower levels N < L** downwardly.*


**Static material causation** occurs at a level L simply through the existence of the materials out of which structures are made. Those materials came into being either naturally through geological processes, as in the case of igneous, sedimentary, and metamorphic rocks, through developmental processes in biology, or artificially through manufacturing processes as in the case of steel, glass, plastics, paper, cloth, and so on (Miodownik 2014 [86]). These materials at level **L** determine what material can exist at levels **M > L**.

**Dynamic material causation** occurs at a level L ***either*** if the set of variables **{X^i^_L_**(**t**)**}** at level **L** at time **t** are altered by same-level interactions at level **L or** if higher-level variables **X^i^_M_**(**t**) (**M > L**) shape a new set of variables **{X^i^_L_^#^**(**t’**)**} = {{X^i^_L_^#^**(**t’**)**, X^i^_M_**(**t’**)**}** at level L; this is ***dynamic downward material causation*** (**DDM**). It takes place by: DDM1:***creating*** or ***importing*** new elements;DDM2:***sustaining*** or ***altering*** elements already in place;DDM3:***deleting*** or ***exporting*** elements.

This downward action to shape the parts to fit higher-level needs is labeled “Machresis” by Gillett (2002) [72]. It leads to the key idea of “*understanding the parts in terms of the whole*” (Cornish Bowden et al., 2004 [158]). There is a key contrast between biology and statistical physics here; nothing like this occurs in the billiard ball model of kinetic theory, which is the paradigm often shaping physicalist thought. However, it occurs in the creation of quasi-particles such as phonons in solid-state physics and Cooper pairs in superconductivity (S Simon 2013 [159], Ellis 2020a [50]).

Dynamic material causation enables a ***contextual branching of dynamics*** through the three DDMn operations listed. The generic form that this takes is: set a selection criterion **C** for desired properties of the variable set **{X^i^_L_**(**t**)**}** at level **L**, where **C** can be a criterion either in terms of the variables at level **L** or in terms of resultant emergent variables at a level **M > L**. Then:IF {the set **{X^i^_L_**(**t**)**}** does not satisfy **C}** THEN {vary **{X^i^_L_**(**t**)**}** to give a new set **{X^i^_L_^#^**(**t’**)**}**)(5)
where the variation is produced by one of {DDM1, DDM2, DDM3}, either in a directed manner or by a randomization process, to provide the new set of variables **{X^i^_L_^#^**(**t’**)**}** to select from. Then:IF {the set **{X^i^_L_^#^**(**t’**)**}** does not satisfy C} THEN {repeat} ELSE {stop}. (6) This is an iterated search procedure in the space of possible variables to find a set of underlying variables that satisfy the selection criteria. It is central in Darwinian evolution (Section 5.5) and in engineering design (Dieter and Schmidt 2021 [48]). Note the role of randomness here: it is a key element in many optimal search procedures (Palmer 2022 [160], Ellis and Noble 2023 [49]). The nature of the variables **{X^i^_L_**(**t**)**}** at every lower level is potentially changed by this mechanism. 

Is material causation always associated with function? Not necessarily; there are a multitude of natural forms out there—planets, rocks, mountains, rivers, lakes, oceans—comprised of many kinds of matter that affect outcomes materially but without associated functions. However, all living systems are associated with functions (Hartwell et al., 1999 [69]), as are all artefacts (Simon 2019 [77]), engineering outcomes (Blockley 2012 [47]), organizations (Etzioni 1964 [141]), and social structures (Elder-Vass 2010 [142]), so material causation in all these contexts will be associated with function.

### 5.2. Material Causation: Materials at the Basis of Engineering, Technology, and Daily Life

Every branch of engineering or technology relies on materials that are shaped by a manufacturing process to in order to play a role in performing some function. These are selected for engineering design (Dieter and Schmidt 2021 [48]) and underlie daily existence (see “*Stuff matters*” Miodownik (2014) [86] for a brilliant description). A key form of material causation is constituted by *purification processes*: centrifuges, filtration, titration, fractionation columns, and so on, carried out in production plants creating pure substrates of a highly specific kind that can then be shaped by manufacturing processes to result in the desired formal causation. This is a central function of chemical engineering and the pharmaceutical industry. Manufacturing processes provide from this basis the components that are then assembled into systems (Dieter and Schmidt 2021 [48]). This involves molding plastics according to specification, weaving, knitting, or crocheting to turn one-dimensional threads into two-dimensional cloth that can then be made into clothes, and so on, providing the components needed for the next level up (Miodownik 2014 [86]). These are all cases of **static material causation** because once the relevant system (a weaving loom, distillery, chemical plant, injection mold) has produced the relevant component, it does not then change it continually in an active way. 

### 5.3. Dynamic Material Causation: Metabolism and the Cardiovascular System

**Metabolism** is the set of reactions whereby organisms are provided with the nutrients they need to survive, converting food to provide energy in the form of ATP and to provide the amino acids, monosaccharides, and nucleotides needed for constructing proteins, carbohydrates, and nucleic acids. Metabolic networks (Jeong et al., 2000 [161]) at the cellular level carry out these processes and so keep cells functioning. They include the citric acid cycle, which oxidizes nutrients to produce usable chemical energy in the form of ATP. They interact with genetic regulatory networks (Goelzer et al., 2008 [162]) in a complex way to respond to changes in environmental conditions (Buescher et al., 2012 [163]). Overall, this is **DDM1:** ongoing dynamic creation of needed lower-level elements (sets of biomolecules) by a next level system (the metabolic network) to meet cellular needs in a contextual way.

**The cardiovascular system**, including the heart and **the lungs** at the physiological systems level, keeps us alive by enabling cells (the basic units of life) to survive. If the system fails, it can cause a heart attack; cells die at the cellular level and damage at the physiological level can be a stroke where parts of the brain become damaged or die. Overall, this is **DDM2:** dynamically sustaining viability of lower-level elements through functioning of higher-level structures.

### 5.4. Dynamic Material Causation: Developmental Processes and Gene Regulatory Networks

Developmental processes produce the hierarchy of cells, tissues, and physiological systems (Figure 1). They have to create the right kinds of cells in the right place at the right time to give the right tissue, and cell types are determined by specific proteins. How are these determined so as to give structure that results in the desired function? This is facilitated by **gene regulatory networks** (Jacob and Monod 1961 [164], Monod et al., 1963 [165], Davidson and Erwin 2006 [166], Karlebach and Shamir 2008 [23]). These networks control the way genes are read at the molecular level to produce specific proteins as needed. This is a form of **DDM1**: creation of proteins that were not there before through the molecular biology of the gene (Watson et al., 2014 [9]) in the context of the cell (Alberts et al., 2015 [19]). **Developmental systems** (Oyama et al., 2001 [40]) are the next level up, using positional information to control the reading of genes via gene regulatory networks and so determining what type of cell should come into being and where: should they be stem cells? bone cells? blood cells? muscle cells? fat cells? nerve cells? and so on. They become these various types of cells by modification of pluripotent cells according to the position in the embryo (Gilbert 2001 [36], Wolpert et al., 2002 [37], Gilbert 2019 [38]). Again, cell shapes and sizes are determined developmentally. This is a form of **DDM2**: modification of the nature of entities that already exist. Then, **apoptosis** is a form of **DDM3**: programmed cell death in multicellular organisms.

### 5.5. Dynamic Material Causation: Adaptive Selection and Evolutionary Processes

Natural selection (Darwin 2019 [33], Dobzhansky 2013 [35]) occurs via replication of living beings with variation of properties at the macrolevel (phenotype) due to variation at the molecular (genotype) level, resulting in different survival and replication probabilities of different phenotypes. Over the course of time, those with a comparative survival advantage tend to replace the others and their genotypes then dominate the gene pool. Thus, natural selection can provide a naturalistic explanation for apparent design (Gardner 2009 [81]). This is a case of downward causation (Campbell 1974 [34]) from the environment to organisms to gene regulatory networks to genes to proteins. This can be considered as either **DDM1**: a process creating genomes that were not there before, or **DDM2**: a process altering genes that are already present in the gene pool. Which is appropriate depends on when one gene is different from another (Kaplow et al., 2022 [167]).

The system **S** is a population, so the statistics of survival and reproduction are key. It is individuals that survive and pass on their genes in a mutated way but it is populations that get selected for at the macrolevel, with this outcome chaining down to the individual level and then the gene level. It is an example of the generic principles of adaptive selection: creating order from disorder according to some selection criterion (see (5) and (6)), with stochasticity/randomness of variation providing the variety to choose from. Mutation and genetic recombination supply the needed genetic diversity. Genotype to phenotype maps for proteins, metabolic networks, and gene regulatory networks result from this process, each selected in an evolutionary way. The huge degeneracy of these maps (Wagner 2014 [11]) is key in explaining how exploration of these spaces can take place in the evolutionary time available. Developmental systems are created by evolution and then in turn shape evolutionary outcomes; overall, it is a circular interactive process because each influences the other, leading to the EVO–DEVO view of how evolution takes place (Gilbert et al., 1996 [39], Carroll 2005, 2008 [41,42], Davidson and Erwin 2006 [166], Collins et al., 2007 [168]). Friston et al. (2023) [169] express it this way:


*“We treat the slow phylogenetic process (natural selection) as furnishing top down constraints (i.e., top-down causation) on fast phenotypic processes (action selection). In turn, the active exchange of the phenotype with its environment provides evidence that is assimilated by natural selection (i.e., bottom-up causation). This multi-scale ontological account is licensed by describing both phylogenetic and phenotypic processes as selecting (extended) genotypes and (extended) phenotypes with the greatest fitness; where fitness is quantified with (free energy) functionals of probability density functions (a functional is a function of a function).”*


A key point resulting from these two comments is the following: 

***Selection shapes all emergent levels L in Figure 1 simultaneously****. It has to do so, because they all work together to enable the organism to function (Noble 2012 [3], Ellis 2020 [117]). Selection is therefore not confined to either the gene level or the organism level: it is a process of selection of effective causation at all emergent levels from the macromolecular level up.* (It does not of course affect the physics or physical chemistry levels. These are determined by the laws of physics.)

This is an extension of Noble’s *principle of biological relativity* (Noble 2012 [3]) from organism structure and function to organism evolution. Thus, physiological systems, tissues, cells, and proteins are all the subject of selection (Wagner 2014 [11]).

Major transitions in evolution (Szathmáry and Maynard Smith 1995 [170]) correspond to changes in the emergent hierarchy, perhaps reflecting new uses of information and higher levels of cooperation through addition of a new emergent level. Thus, ***the hierarchy itself evolves***. 

Overall, the process generically results in better adaptation at the macrolevel, even if that adaptation is hard to see at the molecular level because of the multiple realizability of macrolevel structures at the microlevel. Adaptation can be regarded as organism design (Gardner 2009 [81])—which was Darwin’s great discovery (Darwin 2019 [33], Dobzhansky 2013 [35]). However, genetic drift also takes place: some mutations that do not significantly affect survival rates also occur, so not all emergent features are optimally adapted (e.g., the relation of the optic nerve to the eye). Thus, evolutionary history is embodied in outcomes. 

The same evolutionary principles apply to technologies (Arthur 2009 [2]) and resulting engineering applications (Vogel 2000 [130], Blockley 2012 [47]). Indeed, they apply to design processes in engineering (Dieter and Schmidt 2021 [48]).

## 6. Final Causation

The ultimate driver of all these dynamics is purposeful goals, which is where final causation comes in. Aristotle’s concept of final causation was presented in Section 1.2. This section looks at how an updated version of final causation works out in the context of our present-day understandings of science. It considers an updated version of final causation (Section 6.1), the case of individuals (Section 6.2), the case of organizations and resultant technology (Section 6.3), and the foundations of agency (Section 6.4). 

### 6.1. Updated Version of Final Causation

Essentially, final causation refers to the *reason* that things happen: *Why did it occur*? This is the issue of function or purpose in living beings, individual life, in society and organizations, and hence in technology. The proposed updated version of final causation is:

**Final causation** *occurs when values, purpose, or meaning (“Telos”) shape decisions made by organisms, individuals, groups, or organizations and hence shape their actions, with consequent material outcomes at the macrolevel (personal or organizational) that chain down to affect outcomes at all underlying physical levels.*

In fact, as a referee comments, all living beings have at least an implicit purpose and normativity due to the tautological consequences of natural selection, which gives a value to things that favor persistence.

It is a key form of downward causation, shaping physical outcomes to achieve high-level purposes. The way this happens in the case of individuals is discussed in depth in Noble and Ellis (2022) [114] and in the case of organizations in Carney (2021) [171] and Ellis and Noble (2023) [49]. It may occur through a rational process of meta-analysis (Murphy and Brown 2007 [132]), leading to the view of *The Good Society* (Bellah et al., 1985 [172]) or to be based in narratives embodying values (Kay and King 2020 [161], Johnson et al., 2020 [173]). Again, multiple realizability occurs: there are many ways desired higher-level values can be achieved at lower levels.

As regard to final causation, a referee comments as follows:


*“What is ontologically prior? The metaphysical question is this: are values and meaning just a special cause of final cause (as Aristotle thought), or do they figure in the very definition of final cause (as Ellis proposes)? Which is ontologically prior to which? If Ellis were to embrace Aristotle’s more expansive definition of teleology, this wouldn’t negate any of what he describes about the irreducible role of final causation in the human and social sciences. For Aristotle, the question becomes this: are there real (and even irreducible) powers at the peculiarly human and rational level? If powers exist at that level, like the power of reasoning well or a sensitivity to the real value of actions or states, then final causation also obtains”.*


One could indeed propose this as an alternative to what I suggest here. However, my position is that values are indeed the final cause in human life and hence in technology (Noble and Ellis 2022 [114]). On this view, irreducible powers at the peculiarly human and rational level are the foundation of final causation, enabling it to occur, rather than being final causation itself. 

### 6.2. The Case of Individuals

Final causation in the case of an individual essentially corresponds to self-actualization in Maslow’s hierarchy of needs (Maslow 1943 [174]), updated in Heylighen (1992) [175], and is based in processes of metacognition (Murphy and Brown 2007 [132]), where people have responsibility for creating and sustaining normativity (MacIntyre 1984 [176], Heyes 2022 [177]). There may be major life choices such as to be a doctor or teacher or engineer or artist. Once made, this is the final cause that then shapes all further decisions in a hierarchical fashion: what education is needed to make this possible? Where to obtain that education? How to pay for it?, and so on, reaching down to influencing material outcomes such as molecular changes in a patient’s body or paint being applied to a surface. Final causation may also take the form of issues such as a concern for keeping healthy, chaining down to taking exercise, not smoking, joining a gym, and so on, or wishing to travel to distant places to experience them, resulting in learning a foreign language and booking a ticket. Above all, final causation relates to ethical stances and value choices, where values can be envisaged as lying on a scale from extreme selfishness and self-centeredness on the one hand to extreme generosity and a willingness to be self-sacrificial on the other (Ellis and Noble 2023 [49]). These shape all that one does, including one’s interaction with a community. 

### 6.3. The Case of Organizations and Resultant Technology

All organizations have a purpose, otherwise they would not exist. Problem definition is a complex process (Dieter and Schmidt 2021: p. 10 [48]). Deciding what organizational purpose is, is a key part of management (Drucker 2012 [178]) and leadership (Burns 2004, 2010 [179,180]). The result is a hierarchy of goals (Etzioni 1964: pp. 6–8 [141], Mohr 1973 [181]) that supply the value premises that underlie decisions (Simon 2013 [182] and Scott and Davis 2007: pp. 53–56 [155]). In this hierarchical structure of ends,
*“Each member of a set of behavior alternatives Is weighted in terms of a comprehensive scale of values—the “ultimate” ends”* (Scott and Davis 2007: p. 54 [155]),
which is nothing other than the organizational final cause. One should note here the multidimensional nature of goals: they have a cognitive function, a motivational function, a symbolic function, a justification function for actions taken, and an evaluator function (Scott and Davis 2007: p. 184 [155]). Additionally, they have a normative nature: they relate to what is desirable in a moral sense, which is where transformational leadership arises (Burns 2004, 2010 [179,180]). Note that the fact there is such a moral dimension does not imply that actions of any particular individual, group, or organization will necessarily be morally good, rather that they will lie somewhere on the moral spectrum presented in Ellis and Noble (2023) [49] whether they make this explicit or not, and this will be a key aspect of the final causation taking place. 

Those values, whatever they are, will be changing physical outcomes. They change judicial, political, economic, social, and health outcomes (Etzioni 2010 [183]). Overall, Carney (2021) [171] is a profound examination of final causation in social and organizational contexts. The point then is that this all relates to final causation in technology, for all technology has been developed for some purpose, and will result in real-world occurrences that reflect those purposes to a greater or level degree; the case of digital computers, the internet, and outcomes such as the existence of Amazon to simplify purchasing and social media to enable social contact is a case in point. 

### 6.4. Foundations of Agency and Free Will

Clearly, final causation does not make sense unless we have sufficient agency to make meaningful choices between different the options available to us (Dennett 2015 [184]). 

Strong arguments that this is so include those by Murphy and Brown (2007) [132], Ismael (2016) [185], List (2019) [186], and Potter and Mitchell (2022) [187] (they present a framework of eight criteria that, collectively, describe a system that overcomes the challenges concerning agent causality in an entirely naturalistic and non-mysterious way. The criteria are: (1) thermodynamic autonomy, (2) persistence, (3) endogenous activity, (4) holistic integration, (5) low-level indeterminacy, (6) multiple realizability, (7) historicity, (8) agent-level normativity). Multiple realizability of higher-level states at lower levels plays a key role (List 2019 [186]), as does the fact that we are open systems so it is impossible that detailed knowledge of the state **Σ**: = (**p**_i_,**q**_j_) of every single particle in our bodies at time **t_0_** can determine our future human actions, because those actions are responses to external information **Φ** from the outside world at time **t > t_0_** that is not contained in **Σ**. However, these arguments need to be buttressed by including the way that huge randomness at the molecular level allows downwards selection of outcomes at those levels that fulfil higher-level purposes (Noble and Noble 2018, 2021 [62,188]) in line with the way that molecular machines work (Hoffmann 2012 [70], Brown and Sivak 2019 [189]). It also needs to include the fact of biological emergence of purpose, as Ball (2023: pp. 7, 12) [190] comments:

*“The distinction between physics and biology is sometimes illustrated* via *the thought experiment of repeating Galileo’s (almost certainly apocryphal) Tower of Pisa experiment by dropping a cannonball and a pigeon. To the extent that we can predict what the pigeon will do at all, we implicitly invoke its agency. To explain why it does not simply plummet, it is not enough to invoke aerodynamics; we must also in effect allow that the pigeon does not want to plummet. It manifests its agency by virtue of having goals. … Agency evolves precisely because living organisms are liable to encounter challenges that evolution itself is too slow to adapt to.”*

Our claim now will be:

***Free Will:*** *In the context of the modular hierarchical structure of human life, with multiple realizability of higher-level functions occurring in the context of huge numbers of molecules at the cellular level in a highly dynamic environment (§2.2) and with humans being open systems, dynamic formal and material downward causation underlie the existence of free will.*

This is the essential content of Noble and Noble 2018, 2021 [62,188], Ellis (2021) [191], and Ellis and Noble 2023 [49]. It allows the metacognition and reflection on purpose that is the key element of final causation (Murphy and Brown 2007 [132]). It does not in any way undermine standard physics as set out in Carroll 2022 [192]. 

Physical indeterminacy at the lowest levels is essential to provide some causal slack in the system, whereby macroscopic organization can have any kind of real causal influence over the go of things. There is even literature from Aristotle himself on chance as a “fifth cause”, though it is admittedly confusing and contentious. This counters claims that, for example, the standard model of particle physics is “causally comprehensive” and any talk of causes at higher levels is just a convenient fiction. Since supervenience is merely a matter of a kind of robust correlation, the fact that the emergent properties supervene on the physical base does not mean that the emergent properties are wholly dependent on or wholly grounded by the microphysical. If the emergent properties are truly equally fundamental in our ontology, some special reason must be given for thinking that the emerging properties can vary only if the base properties do. However, the horizontal causal connections between emergent properties can be stochastic or probabilistic, so two worlds can come over time to vary at the emergent level without undergoing any difference at the base level (I thank a referee for the comments in this paragraph). Biology fulfils these conditions because there is major stochasticity at the molecular level (Section 1.4).

## 7. The Outcome

In this section I conclude the discussion of an updated version of Aristotelian causation by considering the four causes and emergence (Section 7.1), a specific example (Section 7.2), causal closure (Section 7.3), and how physics by itself does not give all the answers (Section 7.4).

### 7.1. The Four Causes and Emergence

One can propose that the four causes relate to emergence in the following ways:Stating that efficient causation occurs at each higher level is essentially the claim that emergence does indeed occur. Novelty arises at each higher level because the nature of efficient causation is different at each emergent level.Stating that material causation occurs is essentially stating that supervenience occurs over a material basis.Stating that formal causation occurs is essentially the statement that constraints break symmetries and thereby shape outcomes.Stating that final causation occurs is a statement that in the case of humans and the abstract and material artefacts they create, purpose and values play a key role in determining outcomes.The same level relations between the four causes are that both material and formal causation often play a role in efficient causation at each level. The exception is abstract causation in the case of intelligent beings and organizations.The interlevel relations that relate efficient causation at different levels in biology and technology are mediated by static and dynamic material and formal causation.Final causation plays a key role in determining what happens at all levels in the cases of human beings, organizations, and technology: it, in effect, provides a topmost level of causation via its associated values, as indicated in Figure 1.In each case, outcomes are determined contextually via initial and boundary conditions for the effective laws characterizing efficient causation at each level. The effective laws do not by themselves determine any outcomes at all and that applies inter alia to the efficient causation characterized by physical laws such as Newton’s laws of motion, Maxwell’s equations, and so on. They determine possibilities but not specific outcomes in particular contexts.

### 7.2. A Specific Example

An example of the interactions between the various kinds of causation is given by an example developed from a talk given by Russell Ackoff, namely:


*Why is an aircraft flying from Hamburg to London?*


If carefully considered, it turns out that all forms of Aristotelian causation are involved, as indicated in Table 1.

***Final causation*** arises at **Level L8**, that of the founder and shareholders. They decide the purpose of the company, which is its final cause (cf. Mayer 2018 [193]) from which all else follows. ***Efficient and material causation*** take place at **Level L7**, the level of managers, who timetable flights (efficient causation) and arrange for aircraft and fuel to be available (material causation). Efficient and material causation occur at **Level L6**: the level where engineers design the plane and then manufacture it. A fundamental reason that the plane flies is because it was designed to fly, whereas a house does not fly because it was not so designed. ***Final causation*** occurs again at **Level L5**: the passengers desire the trip because of some final cause or other (family loyalty, a wish to travel, the purpose of a business they run, and so on). ***Efficient causation*** occurs at **Level L4**: the plane flies because the pilot controls the aircraft, opening the throttle, adjusting the height and heading, etc. ***Formal causation*** occurs at **Level L3**, because it is the shape of the wing that makes the plane fly rather than drop out of the sky; much money is spent on researching optimal wing shape. ***Material causation*** also occurs at **Level L3**, because the strength of the wing depends on the materials out of which it is made. ***Efficient causation*** takes place at **Level L2**: Bernoulli’s principle determines the lift that the wing (with its specific shape, so formal causation) will generate when flying at a specific speed, because air molecules are moving faster above the wing than below, and that is based in the kinetic theory of molecular behavior. ***Efficient causation*** at **Level L1** is how the underlying physics enables it all to happen.

All the four forms of causation are at work. The plane will not be flying unless they all are determining outcomes in a coherent way simultaneously. Physical causation is simply one of the many causes entering the picture.

### 7.3. Causal Closure

From this, it follows that causal closure only occurs when all levels linked by upward and downward causation, and the interactions between them occurring via formal and material causation, are taken into account (Ellis 2020 [117]). The plane will not be flying if any of these levels or types of causation are missing because of the interlevel links between them. Thus, it is not true that the physical level by itself is causally closed, as some claim. 

Further examples of causal closure in emergent contexts are, biological organization as closure of constraint (Mossio and Moreno 2010 [194], Montévil and Mossio 2015 [158]); smoking causes lung cancer, driven by advertising and social pressure but arising through molecular biology processes (Ellis and Noble 2023 [49]); determination of values occurs by circular interaction between individuals and society (Ellis and Noble 2023 [49]); the emergence of functionality over time due to natural selection, which is instantiated in some particular molecular configurations of a system meeting specific emergent biological needs, that is, via efficient and material and formal causes. As Aristotle argued, these explanations at different levels and over different timescales are not in contrast with each other—they are all required to fully explain the behavior of the system. A referee argues, 


*“Aristotelians contend, there is no microscopic world that can be exhaustively described in exclusively microscopic terms, because of the irreducible role of macroscopic constraints. This fact renders the usual definition of supervenience (whether global or local) inapplicable, since that definition presupposes that we can disentangle emergent and base properties and facts from each other. Aristotelians should reject the talk of disjoint “levels” of reality in favor of pluralistically-described indivisible whole.”*


My viewpoint, defended above, is that the first two claims here are indeed true and are what is defended in this paper. Where I disagree is the third claim. I agree that the whole acts as an indivisible entity, in the sense that causal closure takes place: outcomes are only determined when all interacting levels are taken into account and in that sense form an integral whole. Nevertheless, that whole does indeed have determinable separate levels, each with real causal powers. This fact is demonstrated by all the examples given here (and is the reason those examples are given). They are solid evidence this claim is true. 

### 7.4. Physics by Itself Does Not Give All the Answers

Greene (2020) [92], Carroll (2021) [93], and Hossenfelder (2022) [94] use titles in articles and books claiming that physics can provide answers to all the biggest questions in the universe. They believe that only efficient causation occurs and that only at some underlying not specified physics level (the problematic issue is that we do not know what the bottom-most physical level is, assuming there is one). Thus, they operate in a framework that does not take into account three of the four Aristotelian causes and takes into account only one level of efficient causation. This viewpoint is unable to explain even so simple an issue as why a teapot exists (Ellis 2005 [195]), nor can it explain neuroscience outcomes (Grasso et al., 2021 [196]). It is the existence of such reductionist viewpoints (Horgan 2023 [197]) that make it important to explicate the various other kinds of causes that demonstrably occur in the real world. This can be systematized by an approach generalizing Aristotle’s causes to be both synchronic and diachronic. 

The ***possibilities*** of biological and technological emergent outcomes are built into the laws of physics. ***Specific outcomes*** depend both on those laws and on contextual and historical factors that are not implied by them. Aristotle’s four causes are all implicated in specific real-world outcomes. Why the laws of physics exist, and why they have the nature they do, is the deep underlying metaphysical question that cannot be answered by science.

## Figures and Tables

**Table 1 entropy-25-01301-t001:** Causal factors when a plane flies from London to Hamburg.

Level	Agent	Causation	Nature
L8	Founder Shareholders	Final	Provide a service; Make a profit
L7	Management	Efficient Material	Timetable flights; Buy aircraft; Arrange fuel supply
L6	Engineers	Efficient, Material	Design the plane; Manufacture it
L5	Passengers	Final	Need for trip
L4	Pilot	Efficient	Control aircraft
L3	Wing	Formal Material	Provides lift; Provides strength
L2	Fluid Flow	Efficient	Bernoulli’s principle
L1	Molecules	Efficient	Kinetic theory

## Data Availability

Not applicable.

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
