# Peer review of "Efficient, Formal, Material, and Final Causes in Biology and Technology"

_entropy, 2023, doi:10.3390/e25091301_

Round 1
Reviewer 1 Report
This is an interesting paper on an important topic, but also slightly frustrating in the way it is put together. The author, George Ellis, has previously made very convincing arguments for the reality and effectiveness of whole-part or top-down causation in complex systems and for the importance of understanding diachronic causal influences in shaping systems architectures and functions, across levels. Here, he aims to modernise Aristotle’s four causes and recruit them in the purpose of defending non-reductive causation or emergentism.
Some questions immediately arise: Is this necessary? Is it helpful? Could we look to modern principles without recourse to Aristotle’s categories? What do we gain from trying to rehabilitate these ancient concepts that we couldn’t get from more general theorising about causation in complex systems?
Aristotle’s causes have an important place in the history of science (especially with Francis Bacon’s admonition to eschew any consideration of formal or final causes – a point that the author might draw out, as it has led explicitly to the materialistic and reductive tendencies of Western science).
I am completely convinced that fully understanding causation and fully explaining why complex systems are the way they are and behave the way they do requires holistic, historical, contextual, extended views of causation across levels. As Aristotle was the first person we know of to articulate this holistic view, it is certainly worth reminding modern readers of his contributions. And the project to relate his four causes to the kinds of causal processes we now understand in complex systems is an interesting one.
However, it must be said that Aristotle’s phrasing or framing of his causes is not easy to parse. Of all the causes, the final cause is the easiest to understand. But the material, efficient, and formal causes are all tied up together in ways that may actively confuse, rather than illuminate. The author thus has a challenge to map these ancient, vague and/or contentious categories to modern scientific concepts in anything more than a loose way.
In particular, if they are to be pressed into service to make the case for holism and against causal reductionism (where all causes come from the bottom up, whatever the “bottom” is taken to be), then the author has to show not just that these concepts can indeed be couched in modern terms, but that this exercise is useful in making or clarifying points that could not be made so clearly otherwise.
Though I am entirely sympathetic to the holistic view and to the goals of this intellectual project, I am not convinced that this case has been made. Nor do I think that more skeptical readers who are inclined towards causal reductionism will have their minds changed by the explication and remapping of Aristotle’s causes presented here.
A major problem in the presentation is that the author simply asserts or stipulates a variety of types of top-down or cross-level causation. There are long lists of instances across various types of systems and scales of reality where such causal influences are referred to. But there is little work done to actually prove, to a skeptical reader, that this is indeed the right way to think about things. The rejoinder could always be that such causal descriptions are merely convenient and that all the real work is still being done at the level of physical forces – it’s just too complicated for us to grasp. The remapping of Aristotle’s notions, while interesting, is unlikely by itself, without some added argument or evidence (such as in Ellis’s prior works), to do the job.
On a stylistic note, the reliance on bullet points and lists makes the paper tough going for the reader. It frankly reads more like an outline than a finished essay in many places. These many, diverse examples are perfectly appropriate to supplement argumentation, but are not a substitute for it.
Ellis writes:
Each emergent level L for a system S is characterized by suitable variables XiL(t) at time t (Anderson 1972). In this context, our updated definition of material causation is, Efficient Causation takes place at an emergent level L in the hierarchy of emergence of a system S when effective laws EL(XiL(t)) at level L determine the values at times t > t0 of the relevant variables XiL(t) at that level from initial data XiL(t0) at time t0. This is a characterization of causation at that level.
This seems to be simply asserting that the causes of things at a given level are indeed variables at that level. The idea that any given level has some causal independence from the goings-on at lower levels relies on ideas of hierarchy, modularity, coarse-graining, multiple realisability, and causal protectorates. Some of these are mentioned, again mostly in bullet points, but they are not really defended, except by reference to the exhaustive list of citations, which places a heavy burden on the reader to do the work of digging out and evaluating the justifications for these ideas.
Where is the proof or argument that causation at some “level” really resides in parameters defined at that level? This is effectively a circular argument – it’s exactly that that makes a level a level. But what justifies it? And what has this definition really to do with Aristotle’s efficient cause?
One interpretation of the efficient cause is that it tries to pick out the most proximal physical impetus of change (or stasis) in some system. In modern parlance, this would seem to map to physical forces. But Ellis seems to want to encompass informational causation, abstract causation, even organisational strategies. It’s not obvious why these should map to efficient causes – indeed, they seem more naturally aligned with formal causes, in that they seem to depend on the organisation or form of things.
Indeed, one general way to reintroduce formal causes into modern parlance is just to recognise that there can be no efficient physical causes without some non-random organisation of stuff – any directionality of physical forces arises from physical inhomogeneities.
With respect to material causes, there is also a multiplication of things that fall under Ellis’s new framing. But, again, it is not so obvious how all these things relate to what Aristotle had in mind. Perhaps the point is to recognise how much more we now know and modernise (and expand) the concept. But if this becomes something so far from the original, then one could be forgiven for questioning what the point of the call-back to the ancient concept is. How does it help us understand our modern complex systems theories?
In particular, Ellis’s reframing of material causation involves these activities:
DDM1: creating or importing new elements; or
DDM2: sustaining or altering elements already in place; or
DDM3: deleting or exporting elements.
To me, these seem to be processes that act on the material and determine what the material make-up of a system will be. But Aristotle’s framing seems to be almost the opposite – more about how the material a system is made of determines what its properties are. (For what it’s worth, there are very interesting perspectives on this latter view in developmental biology, from Stuart Newman, for example, who shows multiple examples of how the tissue structures that form during embryogenesis are constrained and informed by the physical properties of the constituent cells and proteins).
On a side note, I was surprised to see little mention of physical indeterminacy at the lowest levels. By Ellis’s own prior arguments, this seems essential to provide some causal slack in the system whereby macroscopic organisation can have any kind of real causal influence over the go of things. There is even a literature from Aristotle himself on chance as a “fifth cause”, which might be interesting to mention, though it is admittedly confusing and contentious. But some argument around indeterminacy seems necessary to counter claims that, for example, the Standard Model is “causally comprehensive” and any talk of causes at higher levels is just a convenient fiction.
Fred Dretske’s work distinguishing “triggering” and “structuring” causes might also be useful, especially in emphasising the emergence of functionality over time due to selection (final causes), which is instantiated in some particular configurations of a system (efficient and material and formal (organisational) causes). As Aristotle argued, these explanations at different levels and over different timescales are not in contrast with each other – they are all required to fully explain the behavior of the system.
Overall, I am very sympathetic to the project that the author undertakes here. But I feel the execution falls short if the goal is to convince readers who may approach it with a less charitable outlook. The real question for me, which I feel the author does not fully engage with, is what do we gain in our conceptual understanding from this reworking of Aristotle’s causes, that we don’t already have, in different (frankly more familiar) words in the language and science of complex systems?
Author Response
Response to Referee 1
I thank the referee for his comments, and have revised the paper to meet them. Here are my replies to his/her comments
- I Have changed the title and made clear that I am not providing an exegesis of what Aristotle said, but rather using his ideas as a basis for an exploration of emergence in biology and technology
- I have made clear that I am not fighting the battle about strong and weak emergence, but rather simply the claim that emergence does indeed occur.
- “”Where is the proof or argument that causation at some “level” really resides in parameters defined at that level?”” – the examples given, no longer via bullet points, are experimental proof that effective causation indeed occurs at that level. This is not a circular argument, it is a way to test the real world to see how it works and states what the results are. This is the way it is, provided you believe that experimental tests tell us the nature of causation in the real world. It is an established fact, for example, that the beating of a heart keeps a body alive.
Why that is the case is a separate issue from admitting this undeniable fact. It is based in the fact that cells need oxygen to survive – an undeniable fact at a lower level. The fact they are linked does not in any way undermine either statement. As I make clear, I am not trying to establish if this is strong or weak emergence, which is a logically separate issue from the fact that causation at each of these levels works the way it does.
- “”they seem more naturally aligned with formal causes, in that they seem to depend on the organisation or form of things”- an excellent point. I have changed the paper to make clear that efficient causation can be and usually is mediated by both material and formal causation. They are not exclusive but rather co-existent.
- Material causation when generalised can take place in both synchronic form (as Aristotle used it) and diachronic form: how did the stuff get the way it is? Agreed that Aristotle did not make this distinction. This is a generalised form that is useful to us now.
- Why is it worth doing? I claim that my analysis provides a more fine-grained understanding of how same level, upwards, and downwards causation occurs in emergent biological and technological systems than other similar analyses.
- I have made more of a point of physical indeterminacy at the lowest levels than in the previous version, and quoted the referee on this point.
I believe the changes made satisfactorily meet the referee’s comments, and have substantially improved the paper.
Incidentally it says at the top that the referee would like to sign his/her name but nowhere did I see that name.

Reviewer 2 Report
This is an impressive paper. It is extremely well researched and provides a wealth of scientific detail. The main weakness of the paper is philosophical, in that it lacks a clear definition of ‘emergence’. In particular, it is not clear whether Ellis is defending weak or strong emergence—that is, whether the novelty and unpredictability of the emergent properties is merely conceptual, epistemic, or computational (weak emergence) or ontological, irreducible from a God’s eye perspective (strong emergence).
The inclusion of supervenience as an element in the characterization of emergence suggests (although it does not strictly entail) that Ellis has weak emergence in mind. If the emergence were strong, then, as O’Connor and others have pointed out, there is good reason to doubt that the emergent supervenes on any base. It’s true that, since supervenience is merely a matter of a kind of robust correlation, the fact that the emergent properties supervene on the physical base would not entail that the emergent properties are wholly dependent on or wholly grounded by the microphysical. However, if the emergent properties are truly equally fundamental in our ontology, some special reason would have to be given for thinking that the emerging properties can vary only if the base properties do. If, for example, the horizontal causal connections between emergent properties can be stochastic or probabilistic, then two worlds could come over time to vary at the emergent level without undergoing any difference at the base level.
Ellis also fails to distinguish between global or weak supervenience and local or strong supervenience (using Kim’s distinction). Strong or local supervenience is especially hard to reconcile with strong emergence, since the emergent properties should (if they are really metaphysically fundamental) enjoy some measure of independence from the base properties.
Ellis’s account is similar in many ways to that of David Yates. (Yates, David, 2017, “Demystifying Emergence”, Ergo, an Open Access Journal of Philosophy, 3: 809–841. doi:10.3998/ergo.12405314.0003.031) Tim O’Connor’s observation about Yates’s account in the SEP article on Emergence seems apt here:
Yates contends that these examples demonstrate the reality of his distinctive variety of strong emergence. A skeptic might press that the effects Yates cites as pointing to a distinctive kind of higher-level causal power are themselves all higher-level. Assuming that all macroscopic properties are microphysically realized, if one were able to take a wide-angle view of the evolving process in purely micro-physical terms throughout (including in characterizing the targeted token effects), it’s not clear that reference to anything other than the features of and basic relations among microphysical entities is required for explanation. It might well be the case that to explain the token effects under their macroscopic description requires equally macroscopic appeal to molecular geometry (where a given geometric shape is multiple realizable by distinct spatial arrays of atoms). But such explanatory irreducibility is, as we’ve seen, the hallmark of forms of weak emergence.
https://plato.stanford.edu/entries/properties-emergent/#NoveCompPoweConfWhol
An Aristotelian approach to emergence should be helpful at this very point, since the Aristotelian can insist that we cannot fully describe the microscopic level except in terms of the specific ways in which that level is informed and structured by higher levels.
The measurement problem of QM, when approached from a neo-Copenhagen viewpoint, including the contextual emergence model of Ellis-Drossel, fits this Aristotelian picture, since on that model the quantum realm is ontologically incomplete, requiring independent supplementation in terms of irreducible thermodynamic features, like heat baths.
Weak emergentists assumes that we can give an exhaustive description of the world at the microscopic, physical level, thereby fixing with some kind of necessity what higher-order descriptions could be true. But, Aristotelians contend, there is no microscopic world that can be exhaustively described in exclusively microscopic terms, because of the irreducible role of macroscopic constraints. This fact renders the usual definition of supervenience (whether global or local) inapplicable, since that definition presupposes that we can disentangle emergent and base properties and facts from each other. Aristotelians should reject the talk of disjoint ‘levels’ of reality in favor of pluralistically-described indivisible whole.
See for example: Simpson, W.M.R., Horsley, S.A.R. (2022), “Toppling the Pyramids: Physics Without Physical State Monism.” In: Austin, C.J., Marmodoro, A., Roselli, A. (eds) Powers, Time and Free Will. Synthese Library, vol 451. Springer, Cham. https://doi.org/10.1007/978-3-030-92486-7_2
There is one more philosophical issue that could be addressed: the scope of final causation. Ellis proposes that final causation requires the involvement of a mind. This is certainly not Aristotle’s own position. All of the world is characterized by a kind of final or teleological causation. Many recent authors have argued that such universal teleology is an immediate and inevitable consequence of Aristotle’s realism about causal powers. All causal powers and potentialities include an essential reference to possible futures, some of which are never actualized. This “physical” or “natural” intentionality is already a form of final causation---since it means that the thing with a determinate potentiality is “ordered to” the corresponding state under appropriate circumstances. As Thomas Aquinas puts it, “For if an agent were not oriented to some effect, it would not do this more than that.” (Summa Theologiae I-II, Q1, A2)
See also:
Michael Rota, “Causation,” in Oxford Handbook of Aquinas, ed. Brian Davies and Eleonore Stump, Oxford University Press, 2014, pp. 104-14.
Edward Feser, “Teleology: A Shopper’s Guide,” Philosophia Christi 12.1 (2010). 143-59.
George Molnar, Powers: A Study in Metaphysics, Oxford University Press (2003)
UT Place, 1999, p. 227 “Intentionality and the Physical: A Reply to Mumford,” The Philosophical Quarterly 49:225-231.
David Oderberg, “Finality Revived: Powers and Intentionality,” Synthese (2017) 194:2387-2425
The metaphysical question is this: are values and meaning just a special cause of final cause (as Aristotle thought), or do they figure in the very definition of final cause (as Ellis proposes)? Which is ontologically prior to which? If Ellis were to embrace Aristotle’s more expansive definition of teleology, this wouldn’t negate any of what he describes about the irreducible role of final causation in the human and social sciences. For Aristotle, the question becomes this: are there real (and even irreducible) powers at the peculiarly human and rational level? If powers exist at that level, like the power of reasoning well or a sensitivity to the real value of actions or states, then final causation also obtains.
Author Response
Response to Referee 2
I thank the referee for his comments, and have revised the paper to meet them. Here are my replies to his/her comments
- I Have changed the title and made clear that I am not providing an exegesis of what Aristotle said, but rather using his ideas as a basis for an exploration of emergence in biology and technology
- Importantly, I have made clear that I am not fighting the battle about strong and weak emergence, but rather simply the claim that emergence does indeed occur. Much of the referees comments fall away once this is made clear.
- The examples given, no longer via bullet points, are experimental proof that effective causation indeed occurs at that level. This is not a circular argument, it is a way to test the real world to see how it works and states what the results are. This is the way it is, provided you believe that experimental tests tell us the nature of causation in the real world. It is an established fact, for example, that the beating of a heart keeps a body alive.
Why that is the case is a separate issue from admitting this undeniable fact. It is based in the fact that cells need oxygen to survive – an undeniable fact at a lower level. The fact they are linked does not in any way undermine either statement. As I make clear, I am not trying to establish if this is strong or weak emergence, which is a logically separate issue from the fact that causation at each of these levels works the way it does. The comments by Yates quoted by the referred seem inessential agreement with this statement. I have included those comments.
- “”Weak emergentists assumes that we can give an exhaustive description of the world at the microscopic, physical level, thereby fixing with some kind of necessity what higher-order descriptions could be true. But, Aristotelians contend, there is no microscopic world that can be exhaustively described in exclusively microscopic terms, because of the irreducible role of macroscopic constraints. This fact renders the usual definition of supervenience (whether global or local) inapplicable, since that definition presupposes that we can disentangle emergent and base properties and facts from each other. Aristotelians should reject the talk of disjoint ‘levels’ of reality in favor of pluralistically-described indivisible whole.”
I agree with the first two statements but not the third. I have made this clear in the revised paper, where I quote this statement.
- “” measurement problem of QM, when approached from a neo-Copenhagen viewpoint, including the contextual emergence model of Ellis-Drossel, fits this Aristotelian picture”” - Indeed but I decided to focus on biology and technology rather than physics, so as to have a clear target.
- The scope of final causation: First, I have excluded consideration of cosmological context as that raises so many other issues . Second I do not wish to include explicit teleology of any kind as a final cause. The issue of possibilities is very intriguing, but I have chosen to generalise Aristotle’s views in another way, emphasizing how an important proposal is that the ultimate determinant of human and technological outcomes is values (Noble and Ellis 2023). Other possible generalisations could be made. The idea of possibility spaces is key but I think should be the topic of a separate paper.
I believe the changes made satisfactorily meet the referee’s comments, and have substantially improved the paper.
Incidentally it says at the top that the referee would like to sign his/her name but nowhere did I see that name.
Round 2
Reviewer 1 Report
I think the paper has been significantly clarified and the arguments strengthened by the changes made. I note a typo in one of the new sentences:
- It will take generally place via a combination of Formal and Material causation at that level
Page 9 - I presume this should be "generally take"
One final note that the author may wish to revise: I do not think "final causes" should be framed as exclusive to humans, which seems to be indicated (page 19). I would argue that all living beings have at least an implicit purpose and normativity due to the tautological consequences of natural selection, which gives a value to things that favor persistence. By contrast, the current wording might be taken to imply that final causes only correspond to psychologically held goals in human beings. A minor change to caveat the language in this section might dispel that interpretation (presuming the author does not in fact intend it)
Author Response
I thank the referee for these 'helpful comments. I have revised the paper accordingly
Reviewer 2 Report
This version is perfectly acceptable. Ellis has clarified exactly what he intends to claim.
Author Response
I thank the reviewer for his comments. The paper has been greatly improved by them